# Spectral Properties of PMMA Films Doped by Perylene Dyestuffs for Photoselective Greenhouse Cladding Applications

**DOI:** 10.3390/polym11030494

**Published:** 2019-03-14

**Authors:** S. M. El-Bashir, M. S. AlSalhi, F. Al-Faifi, W. K. Alenazi

**Affiliations:** 1Department of Physics & Astronomy, Science College, King Saud University, Riyadh 11461, Saudi Arabia; malsalhi@KSU.EDU.SA (M.S.A.); falfaifi@KSU.EDU.SA (F.A.-F.); walenazi@KSU.EDU.SA (W.K.A.); 2Department of Physics, Faculty of Science, Benha University, Benha 13513, Egypt

**Keywords:** poly(methyl methacrylate), perylene, photosynthesis, photoselective greenhouse claddings

## Abstract

Luminescent polymethylmethacrylate (PMMA) films were prepared by the solvent-casting technique from polymer solution doped with different concentrations of red perylene dyestuffs (KREMER 94720 and KREMER 94739). The effect of the dye concentration on the structure and spectroscopic properties was studied using X-ray diffraction (XRD), transmission electron microscope (TEM) optical absorption, and fluorescence spectroscopy. The optimum dye concentration of photoselective PMMA films was determined by the fluorescence spectroscopy measurements and showed the best emission properties for the doping concentration 10^−3^ wt % of the investigated dyes. The accelerated photostability tests showed promising stability of the prepared films towards terrestrial solar ultraviolet radiation (UVA). The results endorsed a promising application of the investigated films in photoselective greenhouse cladding applications as the optimized film fluoresces at the action spectra of special chlorophyll a.

## 1. Introduction

Polymeric materials, such as polymethylmethacrylate (PMMA), have gained a high interest since they can exhibit unique physical and chemical properties which greatly differ from those of their individual components [1,2,3,4,5,6]. PMMA is the most commercially important acrylic polymer that is sold under several trade names including Perspex^®^, Plexiglas^®^, Lucite^®^, and Acrylite^®^ [7]. It was first synthesized during the 1930s. Due to its high transparency, PMMA is also known as organic glass; this property makes it a perfect alternative to glass for applications that need to be lightweight and have impact resistance [8]. In particular, the photophysical properties of fluorescent dyed PMMA have received considerable attention for interesting applications such as photonics, optoelectronic devices, optical filters, waveguides, luminescent solar concentrators, and greenhouse claddings [9,10,11,12,13]. Perylene diimide (PDI) dyes have drawn attention for many years as efficient dopants for transparent polymers due to their fluorescence properties as industrial pigments [14]. PDI is a polycyclic aromatic hydrocarbon with the chemical formula C_20_H_12_; the dye derivatives of this molecule have been studied for their range of useful properties including high stability, intense absorbance in the UV/visible range, high fluorescence quantum yield up to 100%, and being excellent n-type semiconductors [15,16,17,18,19]. These properties have led to the exploitation of perylenes in various applications such as organic field-effect transistors, thin-film transistors, complex supramolecular systems, and, increasingly, organic photovoltaics (OPVs) [20,21,22]. The recent development of a new class of perylene dyestuffs (KREMER, Berlin, Germany) has renewed the interest of researchers in different fields of industrial applications due to its unique photophysical properties such as long-distance energy transfer when doped in polymeric matrices, the ability to transmit light in the absorption range of the luminescent material, and excellent long-term thermal and photostability [23]. There are specific spectral requirements for the light that enters greenhouses, which is an important factor for each plant. In terms of greenhouse plants, most of them grow best at light wavelengths between 400 and 700 nm [24]. Spectral modifications can be made to affect the quality and quantity of the incoming solar radiation by using particular materials as greenhouse cladding. These claddings are known as “photoselective” as they modify the light spectrum which enters the greenhouse. In this way, photosynthesis and photomorphogenesis and thereby the growth of the plants can be affected [5]. The action spectrum of photosynthesis shows that the green–yellow band is not utilized by chlorophylls; this band of the solar spectrum can be absorbed by the photoselective claddings and fluoresced as a deep red light in order to intensify the irradiance level for the photosynthesis process [9]. There are many requirements for these claddings to enhance the photosynthetic active radiation (PAR), such as long lifetime, mechanical strength, wide absorption, high transmission, and efficient fluorescence that matches the absorption of chlorophyll a [5,9]. In this study, we used PMMA possessing superior mechanical strength, high transmission (93%), and stability against ultraviolet radiation as provided by the manufacturer SABIC^®^. The study aims to optimize the composition and photophysical characteristics of photoselective PMMA films doped with red perylene dyestuffs to increase the plant productivity by harvesting the nonutilized solar spectra inside greenhouses.

## 2. Experimental Techniques

### 2.1. Preparation of Photoselective PMMA Films

Polymethylmethacrylate (PMMA) grains were obtained from SABIC (Riyadh, Saudi Arabia). Fluorescent dyes (KREMER 94720 and KREMER 94739) were purchased from Kremer Pigmente GmbH & Co. (Germany); these dyes are highly stable perylene derivatives specialized for coloring plastics according to the information provided by the supplier. HPLC-grade chloroform (CH_3_Cl) was purchased from Sigma-Aldrich (USA); all materials were used without any further purification. PMMA grains and red fluorescent dyes were dissolved separately in chloroform (CH_3_Cl) and sonicated using magnetic stirrer for 6 h at 40 °C. The polymer solution was doped by different dye concentrations ranging from 10^−5^ to 10^−1^ wt %, sonicated for an additional 6 h, and then cast on highly cleaned glass Petri dishes as illustrated by Figure 1. Fluorescent PMMA films were firstly left to be dried at room temperature, then aged in a drying oven at 50 °C for 6 h to evaporate the residual solvent and cut into 4 cm^2^ rectangular pieces of 50 ± 10 μm thickness as measured by a Fizeau interferometer [25].

### 2.2. Characterization and Measurements

X-ray diffraction (XRD) patterns were obtained by an X-ray diffractometer (SHIMADZU LabXRD-6000, Japan), using CuKα (λ = 1.5406 Å) radiation and a secondary monochromator. The operation voltage of the XRD tube was 30 kV, and the current was 30 mA in the 2θ range (5°–80°) with a 2°/min continuous scan speed. The shape and distribution of the dye molecules were examined by the transmission electron microscope (TEM) (JEOL JEM-1400, Japan). The absorbance of the prepared films was measured at the normal incidence by using a double-beam UV/vis/NIR spectrophotometer (JASCO, model V–770 ST, USA) in the wavelength range of 250–1000 nm. Steady-state fluorescence spectra were recorded in the wavelength range of 200–900 nm using a spectrofluorometer (SHIMADZU, RF-6000 PC, Japan); the spectra were recorded at the excitation wavelengths which were equal to the maximum absorption wavelength of each sample employing the front surface detection (FSD) method [26]. Accelerated photostability test towards UVA radiation had been performed by continuously exposing the samples to a calibrated UV lamp (6 watts, 365 nm wavelength, 115 VAC/60 Hz, Cole Parmer, USA) at room temperature for 24 h, which is equivalent to one year’s exposure to the ultraviolet radiation of the terrestrial solar spectrum.

## 3. Results and Discussion

The distribution of the dye molecules inside PMMA films was studied by TEM microscopy; Figure 2 shows a TEM image of PMMA films doped with 0.1 wt % KREMER 94720. It is clearly noted that the dye molecules are distributed in the form of spherically shaped molecular aggregates having a small diameter in the range of 5–15 nm. This feature was observed for all the prepared films due to the homogeneity of the dopant distribution for the investigated types of perylene derivative dyes.

The structure of photoselective PMMA films was deduced by the XRD diffraction patterns shown in Figure 3. This tool is one of the most effective methods used to conclude the material structure easily. It is observed that all the patterns show the amorphous nature of both pure and dye-doped PMMA as all the patterns are quite similar and exhibit one predominant broad peak; the absence of any sharp peaks in all the XRD patterns revealed that all photoselective PMMA films are amorphous [27]. The observed broad peak is shifted from 22° to 24.5° after doping PMMA with 0.1 wt % perylene dyestuffs; this shift can be attributed to the structural irregularity of the PMMA backbone chain induced by doping [28]. This feature revealed that the strong amorphous nature of PMMA was not affected by the dyeing process as the crystallization is blocked by the dopant molecules in addition to pendent side groups [29].

Figure 4 shows the absorption spectra for the prepared photoselective PMMA films recorded in the wavelength range of 200–1200 nm. The major absorption band appeared in the visible spectrum characterizing the electronic transition S_o_→S_1_ of the dye molecules [9,28]. It is noted that the absorbance values are dramatically increased by increasing the concentration of the investigated perylene dyestuffs; this increase is accompanied by a rich vibrational profile as the spectra became broader around the main absorption peaks, *λ_a(max)_*: 578 nm and 547 nm for perylene dyestuffs KREMER 94720 and KREMER 94739, respectively. The FWHM (Full width at half maximum) of the absorption spectra (Δ*λ_a_*) was determined for all the photoselective PMMA films and listed in Table 1; the values display the remarkable broadening of the spectra achieved by increasing the dye concentrations above 10^−3^ wt %. This behavior can be explained by the formation of dimers and higher molecular weight aggregates recognized by large spectral broadening and obvious deviation from Lambert–Beer’s law [28,30]. Two main types of molecular aggregates can be created by high concentrations of fluorescent dye molecules: J-aggregates, which are side-by-side molecular arrangements that cause bathochromic (red) shift, and H-aggregates, which are face-to-face molecular arrangements that yield hypsochromic (blue) shift [28,31]. The type of molecular aggregation in KREMER 94720 and KREMER 94739 dyes will be confirmed by detecting the spectral shifts in the fluorescence spectra.

The fluorescence spectra for photoselective PMMA films are shown in Figure 5 in the wavelength range of 400–800 nm; both of the investigated dyes fluoresce in the red region of the visible spectrum. The spectra showed a remarkable dependence on the dye concentration since the fluorescence intensity is increased by increasing the dye concentration up to 10^−3^ wt % for the investigated perylene dyestuffs. Generally, at concentrations higher than 10^−3^ wt %, the fluorescence intensity is quenched and red-shifted due to the formation of J-type aggregates [31] which are weakly emissive and cause the quenching of the fluorescence intensity [10,32,33]. This behavior was reported in the literature for other types of perylene dyestuff, in that J-aggregates have absorption and fluorescence wavelengths longer than those of single dye molecules [34].

The absolute fluorescence quantum yield, $\Phi_{f}$, of the investigated films was calculated relative to Rhodamine 101 as mentioned in ref. [35], using the following equation [36,37,38]:(1)$$\Phi_{f}=\Phi_{ref}\;\left(a/a_{ref}\right)\left(n/n_{ref}\right)\left(S_{ref}/S\right)$$
where $\Phi_{ref}$ is the fluorescence quantum yield of the standard reference sample (Rhodamine 101), a is the absorbance, n is the refractive index of the matrix, and S is the area under the fluorescence curve; the values of $\Phi_{f}$ and the fluorescence peak wavelength *λ_f_* are listed in Table 1. It is clear that the highest values of fluorescence efficiency are 94% and 85%, achieved for the films containing 10^−3^ wt % KREMER 94720 and KREMER 94739, respectively. At dye concentrations >10^−3^ wt %, $\Phi_{f}$ is decreased due to the formation of excimers and higher aggregates that have small values of $\Phi_{f}$ [39]. It was reported that the fluorescence weakness is caused by Förster-type energy transfer to the dye aggregates [10,28,35]; the type of aggregates can be determined on the basis of the observed spectral shifts by increasing the doping concentration. In the current case, the fluorescence band arises at lower energy by increasing dye concentration relative to the monomer (single dye molecule) band, termed J-type or Scheibe-type aggregates (fluorescence band shifted bathochromic) [40]. This behavior is well correlated to other reported results for perylene dyestuffs that form J-type aggregates with bent or head-to-tail structures [41,42,43,44,45].

Figure 6 shows the absorption spectrum for chlorophyll a in the red band compared to the absorption and fluorescence spectra of the optimized photoselective PMMA films. It is noticed that the optimized photoselective PMMA films absorb in the band which is not utilized by chlorophylls and re-emit it as red light; this means that the high energy of the UV–visible solar spectrum can be moved to match the irradiance level for the photosynthesis process. The enhancement of the photosynthesis effective light, defined as $\eta_{ph}$, can be empirically calculated by
(2)$$\eta_{ph}=\;T_{f}\;\Phi_{f}\left(\frac{S_{overlap}+S_{ch}\;}{S_{ch}}\right)$$
where $T_{f}$ is the film transmission at the fluorescence wavelength, $S_{ch}$ is the area under the absorption spectrum of chlorophyll, and $S_{overlap}$ is the area of the spectral overlap between chlorophyll absorption and the fluorescence spectra of photoselective PMMA films. The values of $\eta_{ph}$ were calculated and listed in Table 2; these values represent the increase of the effective red band area of the solar spectrum due to the radiative energy transfer. It is noteworthy that the values of $\eta_{ph}$ are well correlated to the fluorescence quantum yield of each film in addition to the effect of the values of $S_{overlap}$. As well, the film containing 10^−3^ wt % KREMER 94720 showed the best properties for greenhouse claddings as it has the closest fluorescence band to the action spectrum of photosynthesis process (chlorophyll a) [10], as well as high fluorescence efficiency. The photostability of the optimized photoselective PMMA films has been tested by calculating the photodegradation (*a_t_/a_o_*), which is the percentage change in the optical absorbance after light exposure for a specific time. Figure 1 shows the photodegradation curves of the optimized photoselective PMMA films exposed to UVA radiation (365 nm) for 24 h. It is noted that two degradation steps can be readily observed obeying the following exponential relation [9,46]:(3)ata0= Cxexp(−Rxt)+a%
where *C_x_* represents the fitting constants, *R_x_* denotes the photodegradation rate constants, and *a_%_* is the percentage change in the absorbance that represents the dye residue after irradiation with UVA for a period *t*. The curves suggested two exponential decays; the first concerns the dye molecules which may exist outside the core of the PMMA free volume, thus the molecules take a short time to photodegrade at fast rate constant *R_1_*. On the other hand, the second rate constant *R_2_* characterizes the slow photodegradation of the caged dye molecules inside the deeper layers of the core formed by three-dimensional chains of the PMMA network [9,46]. The degradation rates were calculated for KREMER 94720 and KREMER 94739, respectively, as depicted in Figure 7; these values show that the degradation rate is decreased to 2.6% and 3.9% of its initial value after 6 h of exposure to UVA radiation. The calculated values of *a%* showed that the absorbance is decreased to 97.5% and 98.5% for KREMER 94720 and KREMER 94739, respectively; this study revealed that both dyes are photostable towards terrestrial solar ultraviolet radiation.

## 4. Conclusions

In the current study, novel photoselective PMMA films were prepared from dye-doped polymer solutions using a low-cost solvent casting technique. The spectral properties were optimized for the application as photoselective greenhouse claddings by controlling the doping concentration. The films showed excellent photophysical properties needed for this application such as broad absorption spectra, high fluorescence efficiency, efficient photosynthetic energy transfer, and photostability. This information was employed to evaluate the enhancement of the photosynthesis effective light, $\eta_{ph}$, that showed its maximum value (89.3%) for PMMA film doped with 10^−3^ wt % KREMER 94720. Additionally, the accelerated photostability tests revealed the long-term stability of the optimized photoselective PMMA greenhouse cladding towards terrestrial solar ultraviolet radiation (UVA). The obtained results have a great economic value for protected cultivation as the photosynthesis process can be accelerated inside greenhouses by harvesting the solar spectrum that is nonutilized by chlorophylls. Extended studies are in progress for evaluating the UV blocking and infrared efficiency of this cladding material, which are very important factors that control the yields and quality of agricultural crops inside greenhouses, especially in hot countries such as KSA.

## Figures and Tables

**Figure 1 polymers-11-00494-f001:**
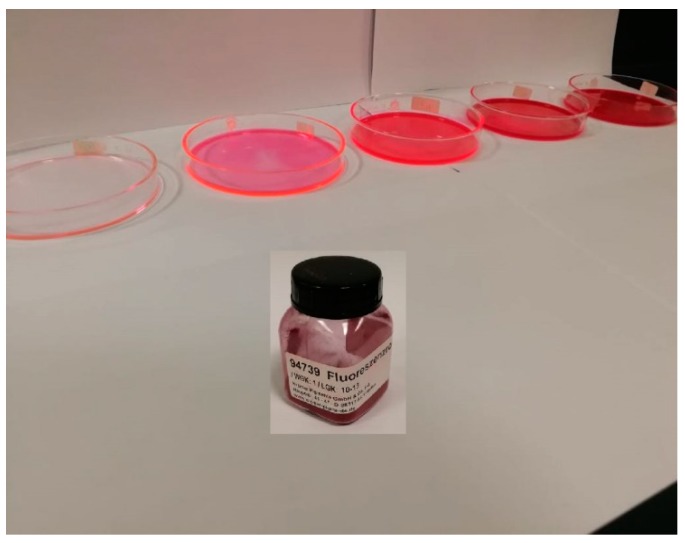
Cast polymethylmethacrylate (PMMA) films from PMMA/KREMER 94739 chloroform solution.

**Figure 2 polymers-11-00494-f002:**
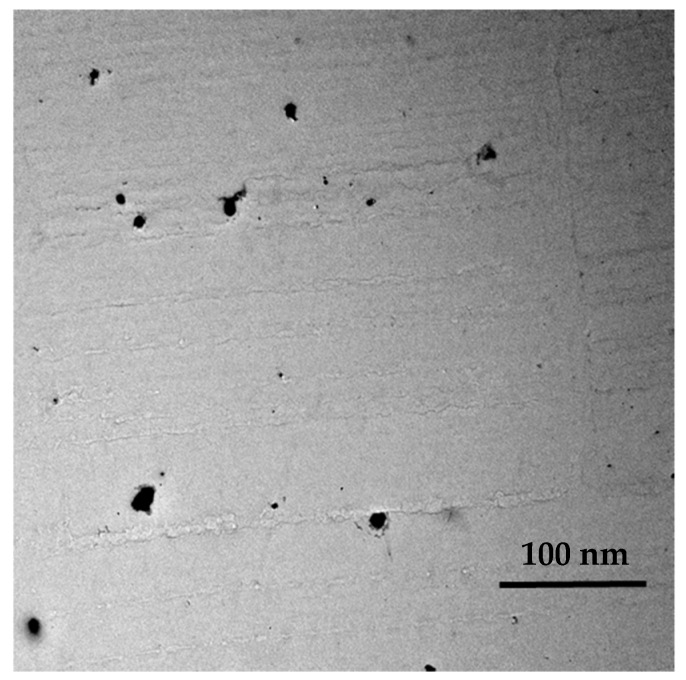
TEM of photoselective PMMA films doped with 0.1 wt % perylene dyestuff (KREMER 94720).

**Figure 3 polymers-11-00494-f003:**
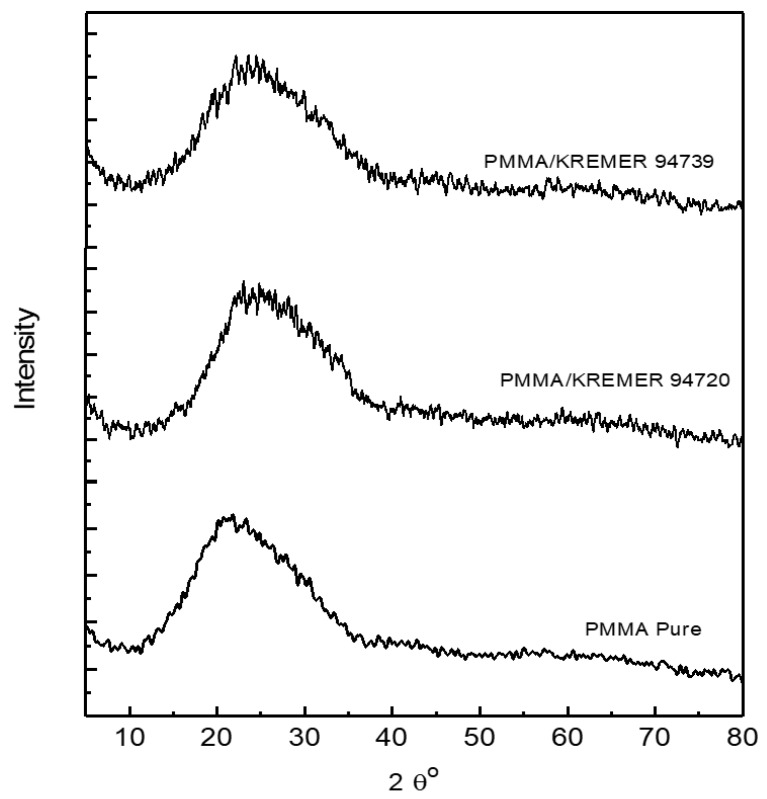
X-ray diffraction of pure and 0.1 wt % dye-doped PMMA films.

**Figure 4 polymers-11-00494-f004:**
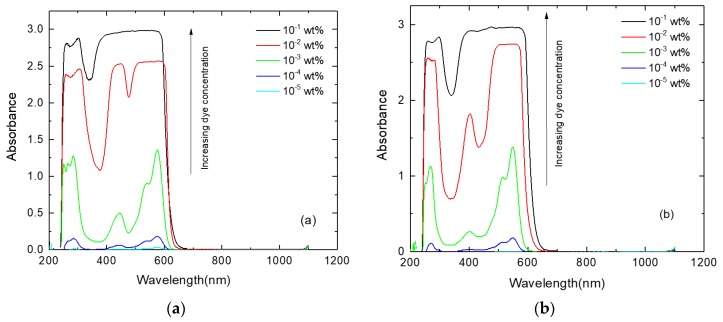
The absorption spectra of photoselective PMMA films doped with perylene dyestuffs (**a**) KREMER 94720 and (**b**) KREMER 94739.

**Figure 5 polymers-11-00494-f005:**
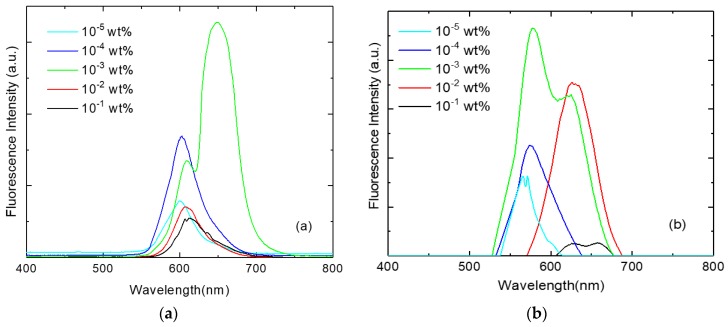
The fluorescence spectra of photoselective PMMA films doped with perylene dyestuffs (**a**) KREMER 94720 and (**b**) KREMER 94739.

**Figure 6 polymers-11-00494-f006:**
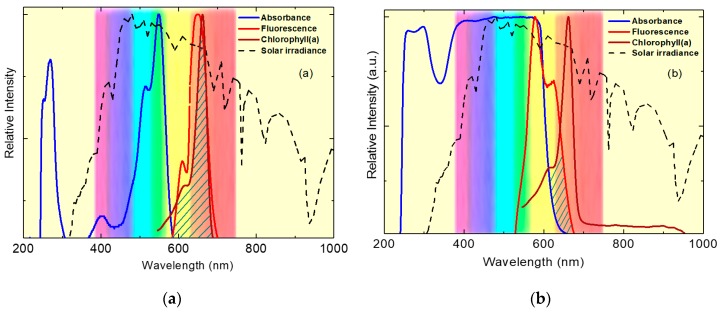
The absorption and fluorescence spectra for photoselective PMMA films doped with perylene dyestuffs compared to the absorption spectrum for chlorophylls: (**a**) PMMA/10^−3^ wt % KREMER 94720 and (**b**) PMMA/10^−3^ wt % KREMER 94739; the dashed area refers to improved light harvesting for the photosynthesis process (all normalized to AM1.5 solar spectrum).

**Figure 7 polymers-11-00494-f007:**
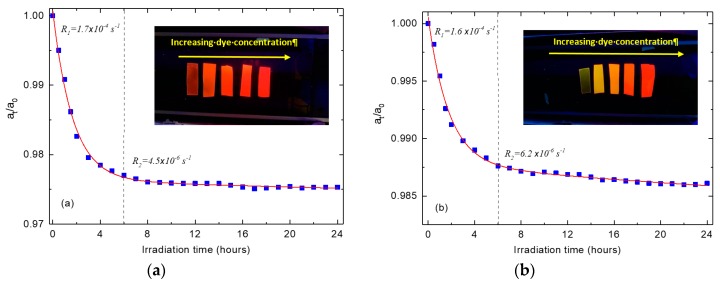
Accelerated photodegradation curves for the optimized photoselective PMMA films: (**a**) PMMA/10^−3^ wt % KREMER 94720 and (**b**) PMMA/10^−3^ wt % KREMER 94739.

**Table 1 polymers-11-00494-t001:** Spectroscopic properties for photoselective PMMA films doped with perylene dyestuffs, FWHM of the absorption spectra Δ*λ_a_*, the fluorescence peak wavelength *λ_f_*, and the fluorescence quantum yield $\Phi_{f}$.

Concentration wt %	KREMER 94720			KREMER 94739		
	Δ*λ_a_* (nm)	*λ_f_* (nm)	$\Phi_{f}$	Δ*λ_a_* (nm)	*λ_f_* (nm)	$\Phi_{f}$
10^−5^	53	584	0.25	51	570	0.29
10^−4^	67	604	0.49	68	574	0.41
10^−3^	72	649	0.94	72	578	0.85
10^−2^	100	608	0.44	82	629	0.64
10^−1^	210	613	0.15	198	644	0.05

**Table 2 polymers-11-00494-t002:** The enhancement of the photosynthesis effective light defined as *η_ph_* for photoselective PMMA films doped with perylene dyestuffs.

Concentration wt %	$\eta_{ph}\;\%$	
	KREMER 94720	KREMER 94720
10^−5^	22.10	5.42
10^−4^	41.37	7.67
10^−3^	89.30	18.70
10^−2^	31.82	11.91
10^−1^	13.40	8.35

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
