# Peer review of "Spectral Properties of PMMA Films Doped by Perylene Dyestuffs for Photoselective Greenhouse Cladding Applications"

_polymers, 2019, doi:10.3390/polym11030494_

Round 1
Reviewer 1 Report
This paper stated that PMMA containing perylene aggregates emits
red light which may be suitable for photosynthesis.
The results and concept were quite clear, so it should be accepted
with minor revision of the following points.
(1) In Fig. 3, difference of XRD patterns were not clear. Please modify
to support your conclusion.
(2) In Fig.4, absorbance may be overflow features. So vibrational profile,
half width, or some other characteristic information mentioned in text
were not clear. Please improve the spectral data to be satisfactory ones.
(3) Around line 112, J- or H-aggregates were mentioned. Usually, molecular
contact may be ristricted in PMMA films. Refering this paper, please explain
molecular-level discussion.
H. Takano, M. Takase, N. Sunaga, M. Ito, T. Akitsu, “Viscosity and intermolecular interaction of organic/inorganic hybrid systems composed of chiral Schiff base Ni(II), Cu(II), Zn(II) complexes having long ligands, azobenzene and PMMA”, Inorganics, 4, 20-29 (2016). doi:10.3390/inorganics4030020
That's all.
Author Response
MANY THANKS TO THE REVIEWER FOR THE USEFUL COMMENTS WHICH WILL STRENGTHEN OUR WORK
1) The results and conclusion part have been improved as suggested and highlighted in the text.
2) We agree with the reviewer opinion that the difference between XRD patterns is not clear, we explained this behavior lines 105-111 as PMMA have a dominant strong amorphous nature which is not affected small dopant concentrations.
3) We clarified in the text that the overflow features are only observed for very high dye concentrations above 10-3 wt% and explained as highlighted lines 123 to 130.
4) Regarding our experience in this field, we found that the discussion in the reference mentioned by the reviewer is not appropriate about dye aggregation because the situation is completely different for metal ions which can not be aggregated as large molecular organic dopants. We cited the proposed reference in the introduction part (ref. 13) as it represents very interesting work about the importance of PMMA in the field of organic-inorganic hybrids.

Reviewer 2 Report
It is an interesting paper and I suggest accepting the paper after minor revision. What I think is missing in the intorduction (or could be done better) is a bit clearer explanation of which polymer properties are needed for geenhouses and why. For instance a high absorption in the wavelength range ... is needed for ... Stability is required because ... etc. In the text I would expect clear references to the points made in the introduction showing how the fabricated films meet the requirements. Concerning the results for instance: what is the relevance of the photoluminescence shown in Fig.5? Also: what light source was used for excitation? Similarly another revision of coclusions would be helpful.
Finally, I would encourage the authors to read the paper again to eliminate any typos.
Author Response
1) The introduction, results and conclusion parts have been improved as suggested and highlighted in the text.
2) The light used for the excitation was equal to the maximum absorption wavelength of each sample as highlighted lines 87-88
WE THANK THE REVIEWER AND APPRECIATE THE USEFUL SUGGESTIONS AND ENCOURAGEMENT.
